

# MicroRNA panel in serum reveals novel diagnostic biomarkers for prostate cancer

Shiyu Zhang[1,*], Cheng Liu[2,*], Xuan Zou[3], Xiangnan Geng[4], Xin Zhou[1], XingChen Fan[1], Danxia Zhu[5], Huo Zhang[6] and Wei Zhu[1]

[1] Department of Oncology, First Affiliated Hospital of Nanjing Medical University, Nanjing, Jiangsu Province, China

[2] Department of Gastroenterology, First Affiliated Hospital of Nanjing Medical University, Nanjing, Jiangsu Province, China

[3] Fudan University Shanghai Cancer Center, Fudan University Shanghai Cancer Center, Shanghai, Shanghai, China

[4] Department of Clinical Engineer, First Affiliated Hospital of Nanjing Medical University, Nanjing, Jiangsu Province, China

[5] Department of Oncology, The Third Affiliated Hospital of Soochow University, Changzhou, Jiangsu Province, China

[6] Department of Oncology, Northern Jiangsu People's Hospital Affiliated to Yangzhou University, Yangzhou, Jiangsu Province, China

[*] These authors contributed equally to this work.

Corresponding authors
Huo Zhang, hollyzh1912@126.com
Wei Zhu, zhuwei@njmu.edu.cn

## ABSTRACT

**Purpose**. MicroRNAs (miRNAs), which could be stably preserved and detected in serum or plasma, could act as biomarkers in cancer diagnosis. Prostate cancer is the second cancer in males for incidence. This study aimed to establish a miRNA panel in peripheral serum which could act as a non-invasive biomarker helping diagnosing PC.
**Methods**. A total of 86 PC patients and 86 normal control serum samples were analyzed through a four-stage experimental process using quantitative real-time polymerase chain reaction. Logistic regression method was used to construct a diagnostic model based on the differentially expressed miRNAs in serum. Receiver operating characteristic curves were constructed to evaluate the diagnostic accuracy. We also compared the 3-miRNA panel with previously reported biomarkers and verified in four public datasets. In addition, the expression characteristics of the identified miRNAs were further explored in tissue and serum exosomes samples.
**Results**. We identified a 3-miRNA signature including up-regulated miR-146a-5p, miR-24-3p and miR-93-5p for PC detection. Areas under the receiver operating characteristic curve of the 3-miRNA panel for the training, testing and external validation phase were 0.819, 0.831 and 0.814, respectively. The identified signature has a very stable diagnostic performance in the large cohorts of four public datasets. Compared with previously identified miRNA biomarkers, the 3-miRNA signature in this study has superior performance in diagnosing PC. What's more, the expression level of miR-93-5p was also elevated in exosomes from PC samples. However, in PC tissues, none of the three miRNAs showed significantly dysregulated expression.
**Conclusions**. We established a three-miRNA panel (miR-146a-5p, miR-24-3p and miR-93-5p) in peripheral serum which could act as a non-invasive biomarker helping diagnosing PC.

# INTRODUCTION

Prostate cancer (PC) is the most frequently diagnosed cancer in men and the second leading cause of male cancer mortality (*Bray et al., 2018*). The vast majority of PC, approximately 95%, presents clinically localized to the prostate without definite metastasis evidence, the 5-year survival of which was almost 100% (*Siegel, Miller & Jemal, 2017*). The initial clinical assessment of the primary tumor has been based on digital rectal examination findings, the serum prostate-specific antigen (PSA) level, and histological confirmation of PC (*Buyyounouski et al., 2017*). However, these methods have some limitations, such as a low predictive value and the adverse consequences of overdiagnosis and overtreatment (*Grossman et al., 2018*). Thus, the exploration of novel noninvasive biomarkers with high sensitivity and specificity for PC diagnosis is urgently needed.

MicroRNAs (miRNAs) are small non-coding RNAs (19–22 nucleotides) that could negatively regulate gene expression at the post-transcriptional level by binding the 3′-UTR of target mRNAs, leading to mRNA degradation and translation repression (*Bartel, 2018*). Several miRNAs have been reported to be stably detected in serum or plasma, suggesting their potential values as new biomarkers for early diagnosis of various kinds of cancers (*Carter et al., 2017*; *Huang et al., 2017*; *Zhang et al., 2019*; *Zhou et al., 2017a*; *Zhou et al., 2015*; *Zhu et al., 2017*; *Wang et al., 2018a*). Meanwhile, researchers found that some miRNAs could discriminate PC patients from normal controls (NCs), thus working as peripheral circulation biomarkers for PC screening (*Shukla et al., 2017*; *Kanwal et al., 2017*; *Moustafa et al., 2017*; *Kumar & Lupold, 2016*). However, the results of researches exploring miRNAs in PC detection were found inconsistent to some extent. There are many reasons for the inconsistency, such as variation of patient characteristics, miRNA analysis techniques, methods of extracting miRNAs, etc. We want to construct a miRNA expression profile of pan-cancer of ourselves.

In the present study, we focused on PC patients and conducted a four-stage study with qRT-PCR to find a potential miRNA profile for detecting PC. We then compared the identified miRNA panel with some other signatures that were previously reported to verify its diagnostic performance. We also did some bioinformatics analysis to confirm our result (*The Gene Ontology Consortium, 2019*; *Kanehisa & Goto, 2000*; *Le, 2019*; *Le et al., 2019*). What is more, the expression of selected miRNAs in tissues and serum exosomes were evaluated to explore their sources and potential forms in circulation. This paper displays preliminary results and will be further verified among larger samples with the more statistical process before clinical application. We hope that our findings may contribute to the construction of promising biomarkers for PC detection.

## MATERIALS AND METHODS

### Study design, patients, and samples

Subjects enrolled in the study were recruited from the Third Affiliated Hospital of Soochow University, Jiangsu Cancer Hospital, and the First Affiliated Hospital of Nanjing Medical University between 2016 and 2017. The Institutional Review Boards approved all procedures of the First Affiliated Hospital of Nanjing Medical University (Ethical Application Ref: 2016-SRFA-149). We obtained the information of TNM stage, differentiation degree, and clinical characteristics from patients' records (according to the seventh edition American Joint Committee on Cancer (AJCC)). All the PC patients were pathologically confirmed. We drew blood samples when patients did not receive any therapeutic procedures like surgery, endocrine therapy, or chemotherapy. NCs enrolled in our study were healthy volunteers who took the routine physical examination at the First Affiliated Hospital of Nanjing Medical University. Each participant provided the written informed consent.

A total of 86 PC patients and 86 normal controls were enrolled in our study. As shown in Fig. 1, we designed a four-stage experimental process. Several factors were taken into account in sample selection and distribution: (i) the purpose of each separate stage; (ii) the sequence of sample collection in practical operation; (iii) the balance of non-experimental factors among four different sets; (iv) the basic principle of experiment design—control, randomization, replication, and balance. Also, factors such as gender, age, TNM stage, and pathological type might have confounding effects on circulating miRNA expression in PC. Therefore, these potential interference factors were as evenly distributed across four sets as possible to avoid selection bias. In the initial screening phase, we randomly chose twenty peripheral serum samples from PC patients, ten from NCs, and then synthesized two PC sample pools and 1 NC sample pool for miRNA profiling (we pooled every ten samples as one pool sample).

Then, to verify the reproducibility and reliability of the results acquired from the screening phase, serum samples from 28 PC patients and 28 NCs were used to confirm the dysregulated miRNAs in the training stage. The miRNAs identified by the training stage were validated in the following testing stage in 32 PC patients and 32 NCs. After the three steps above, we set an external validation phase to further assess the three-miRNA signature's diagnostic value with 26 PC patients and 26 NCs. As shown in Table 1, no significant difference in the distribution of age between PC and NC samples was found in any stage. As previously described in our team's another study (Shan et al., 2018), the identified miRNAs were further validated in 28 pairs of formalin-fixed paraffin-embedded (FFPE) PC tissues and matched adjacent normal tissue specimens from the same surgery patients. Additionally, we analyzed miRNAs' expression in exosomes in 24 PC patients and 24 NCs to investigate the miRNAs' potential form in the peripheral circulation.

We collected 5 ml of venous blood in a vacuum blood collection tube and then put them in a refrigerator at 4 °C. These blood collections were processed within 12 h using a two-step centrifugation method (1,500 rpm*10 min, then 12,000 rpm*2 min) at 4 °C. Next, we carefully aspirated and separated the supernatant to obtain the serum, aliquoted

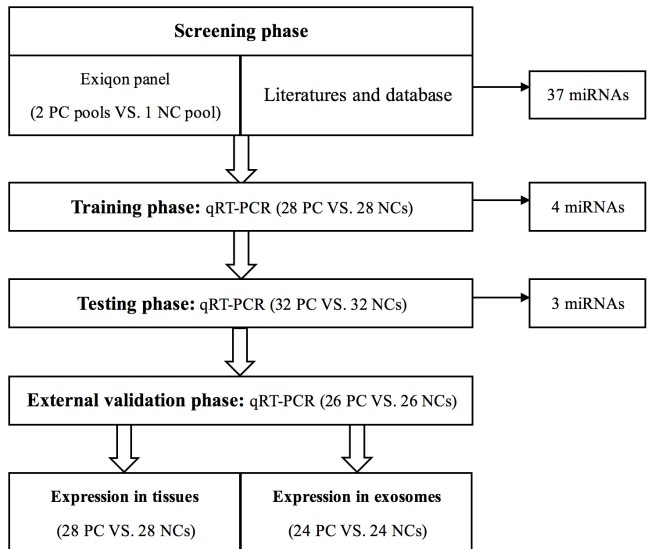

**Figure 1 Overview of the experiment design.** PC, prostate cancer; NC, normal control.

it into clean, enzyme-free EP tubes to prevent repeated freezing and thawing. These serum samples were then stored in a −80 °C refrigerator for further analysis.

## Exosomes isolation

According to the manufacturer's protocol, exosomes were isolated from serum using ExoQuick$^{TM}$ (System Biosciences, Mountain View, Calif). Briefly, 200 µl serum was mixed with 100 µl ExoQuick exosome precipitation solution. Then we precipitated exosome pellets by centrifugation at 13,000 rpm for 2 min. The obtained exosome samples were next lysed into 200 µl RNAse-free water for further analysis.

## RNA extraction

As previously described (*Zhou et al., 2017b*), total RNA was isolated from 200 µl serum or exosome samples using mirVana Paris Kit (Ambion, Austin, TX, USA) under the manufacturer's protocol instruction. We added 5 µl of synthetic C. elegans miR-39 (5nM/L, RiboBio, Guangzhou, China) to each sample after denaturing the solution (Ambion, Austin, TX, USA) for sample-to-sample normalization. We extracted total RNA from FFPE specimens via the High Pure FFPE RNA Micro Kit (Ambion Austin, TX, USA). Then, we dissolved RNA in 100 µl RNAse-free water and stored these RNA samples at −80 °C until use. Nanodrop 2000 Thermo scientific spectrophotometer (NanoDrop Technologies, Wilmington, DE, USA) was used to evaluate the total RNA's concentration and purity.

## Quantitative reverse transcription polymerase chain reaction (qRT-PCR)

The experimental process was mentioned in the previous article (*Zhou et al., 2017b*). MiRNA was reverse transcribed to cDNA using the miRCURY Locked Nucleic Acid

**Table 1  Clinical characteristics of 86 PC patients and 86 healthy controls enrolled in the study.**

| Variables | Training stage ($n = 56$) | | Testing stage ($n = 64$) | | External validation stage ($n = 52$) | |
|---|---|---|---|---|---|---|
| | Cases (%) | Controls (%) | Cases (%) | Controls (%) | Cases (%) | Controls (%) |
| **Number** | 28 | 28 | 32 | 32 | 26 | 26 |
| **Age** | | | | | | |
| <70 | 14 (50.0) | 12 (42.9) | 15 (46.8) | 16 (50.0) | 11 (42.3) | 12 (46.2) |
| ≥70 | 14 (50.0) | 16 (57.1) | 17 (53.1) | 16 (50.0) | 15 (57.7) | 14 (53.8) |
| **Gleason grade** | | | | | | |
| ≤7 | 12 (42.9) | | 18 (56.3) | | 13 (50) | |
| ≥8 | 16 (57.1) | | 14 (43.7) | | 13 (50) | |
| **TNM stage** | | | | | | |
| I | 5 (17.8) | | 7 (21.9) | | 6 (23) | |
| II | 6 (21.4) | | 9 (28.1) | | 5 (19.2) | |
| III | 12 (43) | | 9 (28.1) | | 8 (30.8) | |
| IV | 5 (17.8) | | 7 (21.9) | | 7 (27) | |
| **fPSA\* (ng/ml)** | | | | | | |
| <0.5 | 8(28.6) | | 10 (31.2) | | 8 (30.8) | |
| ≥0.5, < 5 | 6 (21.4) | | 11 (34.4) | | 11 (42.3) | |
| ≥5 | 14 (50) | | 11 (34.4) | | 7 (26.9) | |
| **tPSA\* (ng/ml)** | | | | | | |
| <4 | 4(14.3) | | 5 (15.6) | | 2 (7.6) | |
| ≥4, <40 | 7 (25) | | 14 (43.8) | | 12 (46.2) | |
| ≥40 | 17 (60.7) | | 13 (40.6) | | 12 (46.2) | |
| **fPSA/tPSA** | | | | | | |
| <0.25 | 25 (89.3) | | 29(90.6) | | 25 (96.2) | |
| ≥0.25 | 3 (10.7) | | 3(9.4) | | 1 (3.8) | |

**Notes.**
fPSA\*:, free prostate-specific antigen; tPSA\*:, total prostate-specific antigen.

(LNATM) Universal Reverse Transcription (RT) microRNA PCR, Polyadenylation, and cDNA synthesis kit (Exiqon miRNA qPCR panel; Vedbaek, Denmark). We then scanned the microarrays on a 7900HT real-time PCR system (Applied Biosystems, Foster City, CA, USA) with Exiqon miRCURY-Ready-to-Use-PCR-Human-panel-I + II-V1.M (Exiqon miRNA qPCR panel; Vedbaek, Denmark). The amplification of miRNA was performed via specific primers of reverse transcription (RT) and polymerase chain reaction (PCR) (RiboBio, Guangzhou, China). The RT reaction was performed at 42 °C for 60 min followed by 70 °C for 10 min; the PCR reaction was performed at 95 °C for 20 s, followed by 40 cycles of 95 °C for 10 s, 60 °C for 20 s and then 70 °C for 10 s on a 7900HT real-time PCR system. qRT-PCR was performed to amplify and detect the miRNAs with SYBR Green Dye (SYBR® Premix Ex TaqTM II; TaKaRa, Dalian, China). The expression of miRNAs in serum samples and exosomes were calculated using the comparative $2^{-\Delta\Delta Ct}$ method normalized to exogenous reference miRNA (cel-miR-103a-3p), $\Delta Ct = Ct_{miRNA} - Ct_{cel-miR-103a-3p}$. Tissue miRNAs were determined by the $2^{-\Delta\Delta Ct}$ method relative to cel-miR-39 and RNU6B (U6) (*Livak & Schmittgen, 2001*).

## Statistical analysis

Mann–Whitney test was used to analyze the differential expression of miRNAs between PC patients and NCs. Moreover, the association between miRNAs and the clinical characteristics was estimated using one-way ANOVA or $\chi^2$ test. Sample size calculation has been done using software PASS. A sample of 12 from the positive group and 12 from the negative group achieves 91% power to detect a difference of 0.3 between the area under the ROC curve (AUC) under the null hypothesis of 0.5 and an AUC under the alternative hypothesis of 0.8 using a two-sided z-test at a significance level of 0.15. The receiver operating characteristic (ROC) curves and the area under the ROC curve (AUC) calculation, along with Cox's regression models, were applied to estimate the diagnostic and prognostic value of identified miRNAs. A logistic regression model for PC prediction was used to the data of the training and validation stages. SPSS (version 19.0, IBM, Armonk, New York, USA) and GraphPad Prism 7 (GraphPad Software, USA) were used to performed statistical analyses. Differences were considered significant when $P$-value < 0.05.

## Bioinformatics analysis

Pathway enrichment analyses of miRNA gene targets and differentially expressed genes were performed using DIANA-mirPath v.3.0 (http://www.microrna.gr/miRPathv3) (*Vlachos et al., 2015*). We conducted Gene Ontology (GO) (*The Gene Ontology Consortium, 2019*) analysis and Kyoto Encyclopedia of Genes and Genomes (KEGG) (*Kanehisa & Goto, 2000*) analysis to predict possible biological processes and potential signal pathways. To further confirm our results, we downloaded prostate adenocarcinoma data from The Cancer Genome Atlas (TCGA) and performed a similar analysis.

# RESULTS

## Discovery of candidate miRNAs from the screening phase

To filter candidate miRNAs in the screening phase, we screen the expression levels of 168 miRNAs in the 2 PC sample pools and the NC sample pool using Exiqon miRCURY-Ready-to-Use-PCR-Human-panel-I + II-V1.M on the qRT-PCR platform. Screening criteria for candidate miRNAs were as previously described (*Zhou et al., 2017b*). Only the miRNAs with cycle threshold (Ct) values <37 and 5 lower than negative control in the panel were eligible for further analysis. A miRNA was considered as a candidate miRNA if it showed more than 1.5-fold or less than 0.67-fold altered expression in both two pooled PC samples compared to the NC pool sample. As shown in Table 2, a total of 37 miRNAs was found to exhibit dysregulated expression in PC and chosen for the further training stage.

## Confirmation of identified miRNAs by qRT-PCR

The expression of the 37 candidate miRNAs selected via the screening phase was further analyzed in 28 PC patients and 28 NCs using qRT-PCR in the training phase. 4 of the 37 miRNAs were further confirmed. During the testing stage, all four miRNAs were subsequently analyzed in 32 PC patient samples and 32 NCs. 3 of the 4 show consistently up-regulated expression and met our standards (Table 3). We then assessed the three miRNAs (miR-146a-5p, miR-24-3p, and miR-93-5p) in 26 PC patient samples and 26 NCs

**Table 2** MiRNAs that are differently expressed in the two PC pools in the screening phase.

| MiRNA | Fold change | | Mean fold |
|---|---|---|---|
| | Pool 1 | Pool 2 | |
| hsa-let-7g-5p | 1.562103 | 1.516144 | 1.5391235 |
| hsa-miR-1-3p | 1.592673 | 3.164749 | 2.378711 |
| hsa-miR-10b-5p | 1.753225 | 11.63021 | 6.6917175 |
| hsa-miR-126-3p | 1.888629 | 2.330068 | 2.1093485 |
| hsa-miR-144-5p | 2.786662 | 3.532729 | 3.1596955 |
| hsa-miR-152-3p | 1.808996 | 1.756634 | 1.782815 |
| hsa-miR-155-5p | 4.610215 | 3.432634 | 4.0214245 |
| hsa-miR-15b-3p | 1.727401 | 1.952506 | 1.8399535 |
| hsa-miR-195-5p | 5.29328 | 3.112925 | 4.2031025 |
| hsa-miR-28-5p | 2.37564 | 2.142608 | 2.259124 |
| hsa-miR-296-5p | 1.838551 | 2.485472 | 2.1620115 |
| hsa-miR-29a-5p | 4.160826 | 2.80201 | 3.481418 |
| hsa-miR-30a-5p | 1.511533 | 1.824362 | 1.6679475 |
| hsa-miR-30e−3p | 1.563817 | 2.235089 | 1.899453 |
| hsa-miR-335-5p | 2.034361 | 2.060766 | 2.0475635 |
| hsa-miR-551b-3p | 1.818341 | 8.612135 | 5.215238 |
| hsa-let-7a-5p | 1.53924 | 1.680264 | 1.609752 |
| hsa-miR-103a-3p | −3.57075 | −4.40626 | −3.988505 |
| has-miR-106a-5p | 2.061031 | 2.266201 | 2.163616 |
| hsa-miR-141-3p | −44.0872 | −36.1883 | −40.13775 |
| hsa-miR-16-5p | 1.60983 | 1.780559 | 1.6951945 |
| hsa-miR-21-5p | 2.174586 | 1.516818 | 1.845702 |
| hsa-miR-210-3p | 1.577939 | 2.310396 | 1.9441675 |
| hsa-miR-223-3p | 1.674394 | −11.1606 | −4.743103 |
| hsa-miR-24-3p | 1.585616 | 1.875112 | 1.730364 |
| hsa-miR-26b-5p | 1.554088 | −2.66348 | −0.554696 |
| hsa-miR-30c-5pc | −2.05815 | −2.29057 | −2.17436 |
| hsa-miR-375-3p | −1.78726 | −1.81663 | −1.801945 |
| hsa-miR-93-5p | 1.510669 | 1.892198 | 1.7014335 |
| hsa-miR-146a-5p | 1.53914 | 1.651484 | 1.595312 |
| hsa-miR-200a-3p | 6.006406 | 4.104479 | 5.0554425 |
| has-miR-200c-3p | −2.97765 | −1.70005 | −2.33885 |
| has-miR-574-3p | −4.33151 | −6.01175 | −5.17163 |
| hsa-miR-107 | −5.16725 | −6.01989 | −5.59357 |
| hsa-miR-133b | −1.82926 | −1.64685 | −1.738055 |
| hsa-miR-143-3p | −2.76443 | −3.21305 | −2.98874 |
| hsa-miR-221-3p | −2.33089 | −1.75637 | −2.04363 |

**Notes.**
Each ten samples are mixed as one pool sample.

**Table 3  Expression levels of the three serum mi-RNAs in the training and testing phases.**

| miRNA | Training phase | | | | Testing phase | | | | Combined | |
|---|---|---|---|---|---|---|---|---|---|---|
| | PC | NC | FC | *P* value | PC | NC | FC | *P* value | FC* | *P* value |
| miR-146a-5p | $-3.73 \pm 1.26$ | $-2.99 \pm 1.18$ | 1.68 | 0.031 | $-3.15 \pm 1$ | $-1.81 \pm 1.15$ | 2.53 | <0.001 | 2.24 | <0.001 |
| miR-24-3p | $-2.73 \pm 1.11$ | $-2.01 \pm 0.86$ | 1.64 | 0.014 | $-3.40 \pm 0.98$ | $-2.50 \pm 0.6$ | 1.88 | <0.001 | 1.81 | <0.001 |
| miR-93-5p | $-1.93 \pm 1.36$ | $-0.86 \pm 0.76$ | 2.1 | 0.002 | $-1.92 \pm 1.11$ | $-0.74 \pm 1.02$ | 2.27 | <0.001 | 2.22 | <0.001 |
| miR-143-3p | $0.44 \pm 1.4$ | $1.05 \pm 1.37$ | 1.53 | 0.016 | $0.67 \pm 0.96$ | $1.24 \pm 0.89$ | 1.48 | 0.028 | | □ |

Notes.

FC*, fold change. (presented as mean $\pm$ SD; $\Delta$CT: relative to miR-103a-3p).

The combined column stands for a combination of the two cohorts (the training and testing stages).

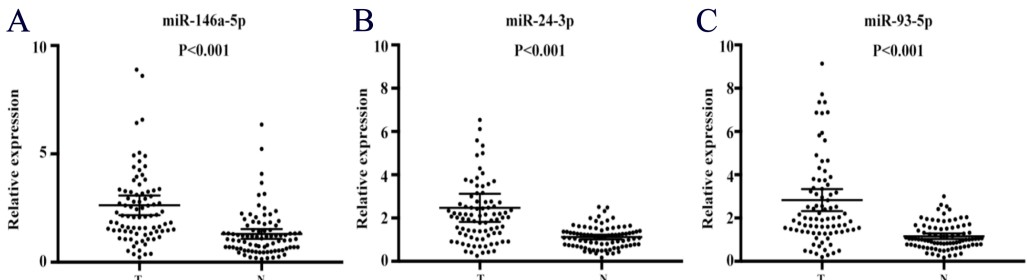

**Figure 2  Expression on levels of three miRNAs in serum of 60 PC patients and 60 controls (in the training and testing stages).** (A) miR-146a-5p. (B) miR-24-3p. (C) miR-93-5p. PC, prostate cancer; T, tumor; N, normal control. Horizontal line: mean with 95% CI.

in an external validation stage and found that the result was the same as above. What's more, all the three miRNAs had significantly higher expression levels in peripheral serum of PC patients as compared with NCs when the results of the three stages were combined (Fig. 2).

## Diagnostic value of the candidate miRNAs

To evaluate the three miRNAs' performance in PC diagnosis, we conducted ROC curves on each of the three miRNAs. Taken three phases into consideration (training phase, testing phase and external validation phase), the AUC for serum miR-146a-5p, miR-24-3p and miR-93-5p was 0.767 (95% confidence interval (CI): 0.696–0.839), 0.767 (95% CI [0.694–0.840]) and 0.781 (95% CI [0.710–0.852]), respectively (Fig. S1). Therefore, each of these three miRNAs had the ability to distinguish PC patients from healthy controls. In addition, when these three miRNAs were treated as a whole, we found that this 3-miRNA signature had a better diagnostic performance. As shown in Fig. 3A, combining the three stages, the AUC of this miRNA panel was 0.819 (95% CI [0.754–0.885]; sensitivity 75.6%, specificity 85.7%). Meanwhile, the diagnostic value of the 3-miRNA panel was assessed in a combination of the former two cohorts (the training and testing stage). The AUC was 0.831 (95% CI [0.757–0.905]; sensitivity = 73.3%, specificity = 80.0%; Fig. 3B). In the external validation stage, similar to the previous training and testing phases, the three identified miRNAs showed a consistent tendency of upregulation, and the panel still performed quite

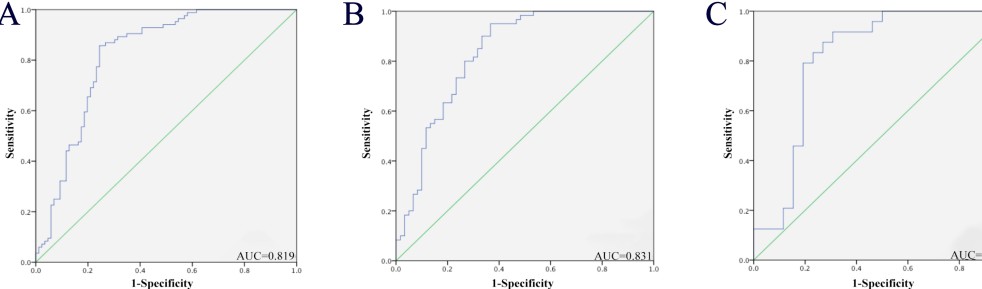

**Figure 3** **Receiver-operating characteristic (ROC) curve analyses of the three-miRNA signature to discriminate PC patients from normal controls.** (A) The combined three cohorts of training, testing, and validation stages (86 PC vs. 86 NCs). (B) The combination of the former two cohorts (the training and testing stage; 60 PC vs. 60 NCs). (C) External validation stage (26 PC vs. 26 NCs). PC, prostate cancer; NC, normal control; AUC, areas under the curve.

well in discriminating PC patients from NCs with the AUC being AUC of 0.814 (95% CI [0.687–0.942]; sensitivity = 76.9% and specificity = 83.3%; Fig. 3C). The correlation between the serum PSA values, TNM stages, and the three serum miRNAs levels was also analyzed, respectively. However, no significant results were observed. Therefore, data were not presented.

## External validation in public databases
We further verified the diagnostic value of the 3-miRNA signature in four public datasets: The Cancer Genome Atlas Prostate Adenocarcinoma (TCGA-PRAD) dataset (http://cancergenome.nih.gov/), the GSE113740 dataset, the GSE113486 dataset, and the GSE112264 dataset. The first TCGA-PRAD dataset represents data from tissues, while the other three datasets contain data of serum samples. In the TCGA-PRAD dataset, the 3-miRNA marker performed well in distinguishing between PC tumor tissues ($n = 498$) and normal tissues ($n = 51$). As shown in supplementary Table S1, the corresponding AUC of the 3-miRNA signature was 0.979 (95% CI [0.967–0.991], $P < 0.001$, specificity: 94%, sensitivity: 94.1%). While in the three datasets of serum samples (GSE113740, GSE113486 and GSE112264), the AUC of the 3-miRNA panel to distinguish PC patients from healthy people was 0.806 (95% CI [0.733–0.880], $P < 0.001$, specificity:68.7%, sensitivity:84.0%), 0.943 (95% CI [0.901–0.986], $P < 0.001$, specificity: 100%, sensitivity: 42.5%) and 0.950 (95% CI [0.925–0.976], $P < 0.001$, specificity: 92.7%, sensitivity: 86.2%), respectively (Fig. 4). To summarize, the selected miRNA panel's diagnostic value was successfully verified in the four public datasets.

## Comparison with previously identified signatures
To further verify the diagnostic capabilities of the 3-miRNA marker, we collected six types of miRNA signatures in serum reported by previous studies and then conducted their ROC curve analysis in the four public datasets mentioned above. As shown in Table S1 and Fig. S2, the 3-miRNA signature has the highest AUC value in the TCGA-PRAD dataset compared to other miRNA panels. In the three GEO datasets, the performance

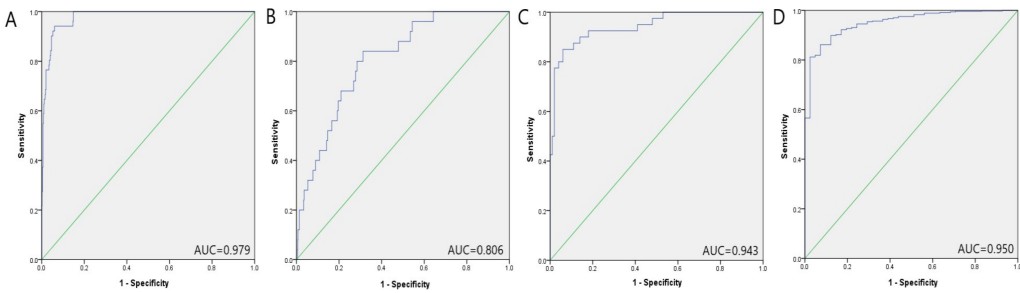

**Figure 4  Prostate cancer diagnostic capability of the three-miRNA panel verified in the four public datasets.** (A) TCGA: AUC = 0.979, 95% CI [0.967–0.991], $P < 0.001$, Sensitivity = 94.1%, Specificity = 94.0%.(B) GSE113740: AUC = 0.806, 95% CI [0.733–0.880], $P < 0.001$, Sensitivity = 84.0%, Specificity = 68.7%.(C) GSE113486: AUC = 0.943, 95% CI [0.901–0.986], $P < 0.001$, Sensitivity = 42.5%, Specificity = 100.0%.(D) GSE112264: AUC = 0.950, 95% CI [0.925–0.976], $P < 0.001$, Sensitivity = 86.2%, Specificity = 92.7%.

of the 3-miRNA panel was only behind the 12-miRNA signature proposed by Bryant et al.. In conclusion, the 3-miRNA panel selected in this study is superior to other miRNA biomarkers in the diagnosis of PC (*Porzycki et al., 2018*; *Moltzahn et al., 2011*; *Chen et al., 2012*; *Bryant et al., 2012*; *Yaman Agaoglu et al., 2011*; *Paunescu et al., 2019*).

## Prognostic value of the three miRNAs for PC

Using data from the TCGA-PRAD dataset, we constructed Cox regression models to estimate the correlation of several influence factors and overall survival (OS) and the three serum miRNAs' prognostic value for PC. However, none of the three miRNAs were significantly associated with prognosis with $P > 0.05$ (Table S2).

## Expression of the miRNAs in tissue samples

We next detected the expression levels of the three miRNAs in 28 pairs of tissue samples by qRT-PCR. As shown in Fig.S3, the results were not statistically significant in this stage. However, when we analyzed the three miRNAs in the un-matched tissue samples with larger sample size through The Cancer Genome Atlas (TCGA), as shown in Fig. 5, miR-146a-5p and miR-93-5p were found to be consistent with those in serum. In contrast, miR-24-3p was found down-regulated compared with NCs, which was opposite to the expression level of miR-24-3p in serum.

## Expression of the identified miRNAs in serum exosomes

Additionally, the expression levels of miRNAs in exosomes were assessed in 24 PC patient samples and 24 NCs by qRT-PCR to explore the potential existing form of the identified miRNAs in PC patients' serum. Compared to NCs, all the three miRNAs were up-regulated, but only miR-93-5p was statistically significant in serum exosomes (Fig. 6).

## Bioinformatics analysis of the identified miRNAs

We assessed these experimentally identified miRNAs with DIANA-miRPathv3.0, a web-server aiming for pathway analysis. By performing the Kyoto Encyclopedia of Genes and Genomes (KEGG) and Gene Ontology (GO) analyses, we analyzed the validated miRNAs'

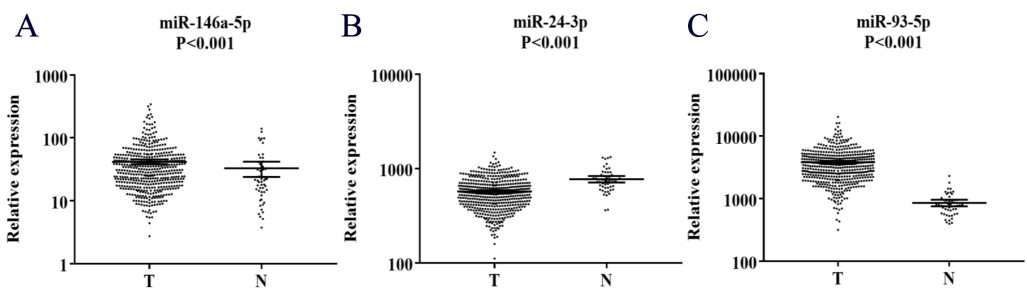

**Figure 5  Expression of the three miRNAs (data from TCGA).** (A) miR-146a-5p. (B) miR-24-3p. (C) miR-93-5p. T, tumor; N, normal control; TCGA, The Cancer Genome Atlas.

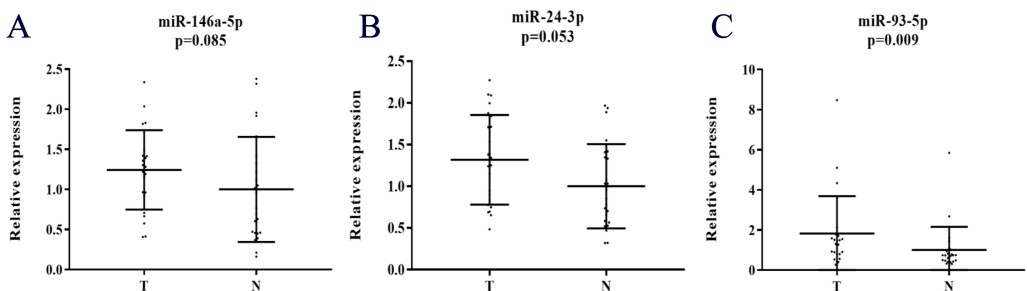

**Figure 6  Expression of the three miRNAs in the serum exosome samples of 24 PC patients and 24 NCs.** (A) miR-146a-5p. (B) miR-24-3p. (C) miR-93-5p. T, tumor; N, normal control.

potential roles that might take part. As shown in Fig.S4A, KEGG analysis showed that these identified miRNAs might participate in some pathways such as AMPK signaling pathway, pathway in cancer, MAPK signaling pathway, TGF-beta signaling pathway, p53 signaling pathway, and so on. As demonstrated in Fig.S4B, GO analysis showed that in RNA metabolic activity, cell cycle, cell proliferation, cell death, and many other biological processes, the three miRNAs should not be ignored.

## DISCUSSION

In the present study, we carefully designed a procedure to identify serum miRNAs with potential for PC detection. In the screening stage, we analyzed differential expression profiles of miRNAs in 2 PC and 1 NC serum pools using Exiqon miRNA qPCR Panels, which showed more sensitivity and linear compared to the TaqMan platform (*Steffen et al., 2011*). Then, we conducted three phases for validation to control the false-positive rate (the training stage, the testing stage, and the external validation stage) based on qRT-PCR subsequently. Ultimately, based on a cohort of 172 serum samples, including 86 PC patients and 86 NCs, three up-regulated serum miRNAs (miR-146a-5p, miR-24-3p, and miR-93-5p) were identified, which simultaneously proved to have a high accuracy in PC diagnosis. Compared with a single miRNA, the molecular signature of the 3-miRNA combination could provide more reliable information about the disease status, and this

expression profile might become a diagnostic molecular signature for prostate cancer in the future.

To further confirm the diagnostic value of the 3-miRNA label, we checked previously published articles that similarly identify serum miRNAs as diagnostic biomarkers for PC. It's important to note that these biomarkers should distinguish PC patients from healthy people rather than differentiate tumor tissue from normal tissue. Moreover, only serum miRNAs were included for the analysis to ensure comparability. Several studies have identified circulating miRNAs as promising biomarkers in the diagnosis of PC (*Kanwal et al., 2017*; *Moustafa et al., 2017*; *Kumar & Lupold, 2016*; *Daniel et al., 2017*; *Cochetti et al., 2016*; *Endzeliņš et al., 2016*). However, we could see that conclusions showed inconsistent results from variations of patient characteristics, techniques for miRNA analysis, and methods of extracting miRNAs (*Endzeliņš et al., 2016*). While incorporating this 3-miRNA signature with some of the previously identified markers may yield a higher sensitivity or specificity for PC diagnosis, most of these miRNAs were not detected by qRT-PCR in our study because they failed to meet the criteria in the screening stage.

By consulting the previous literature, we learned the relevant information about these three miRNAs. MiR-146a-5p played a vital role in the process of androgen-independent prostate cancer cell apoptosis by targeting the ROCK/Caspase 3 pathway (*Xu et al., 2015*). *Sun et al. (2014)* suggested that miR-146a-5p acted as a cancer suppressor through down-regulation of Rac1. Meanwhile, *Daniel et al. (2017)* identified another miRNA panel as a diagnostic biomarker, including miR-146a-5p, which confirmed our current research to a certain extent. Circulating miR-146a-5p was reported to show dysregulation in various types of tumors, such as oral squamous cell carcinoma (*Min et al., 2017*), breast cancer (*Si, Yu & Yao, 2018*), non-small cell lung cancer (*Yuwen et al., 2017*), bladder cancer, and many other types of tumor (*Iacona & Lutz, 2019*). *Fredsøe et al. (2019)* suggested that the up-regulated expression of miR-24-3p in urine made it possible to be a biomarker for PC detection. *Cochetti et al. (2016)* also analyzed circulating miR-24-3p between PC patients and people with benign prostatic hyperplasia but came to a negative conclusion. There were too many other studies supporting that miR-24-3p functioned in various types of tumor except for PC. *Kang et al. (2017)* suggested that miR-24-3p could function as a tumor suppressor by targeting p130Cas in metastatic cancer. *Gao et al. (2015)* revealed that over-expression of miR-24-3p could suppress colorectal cancer cell proliferation, cell migration, and invasion. However, reports also showed that miR-24-3p might promote cell migration and proliferation in lung cancer (*Yan et al., 2018*) and bladder cancer (*Yu, Jia & Dou, 2017*). *Khodadadi-Jamayran et al. (2018)* found that patients with breast cancer, compared to those with low miR-24-3p levels, patients with the primarily high expression level miR-24-3p seemed to have a lower survival rate. As for head and neck squamous cell carcinoma, *Sun et al. (2016)* believe that miR-24-3p could promote cell proliferation and regulate chemosensitivity. What's more, *Meng, Wang & Jia (2014)* suggested that miR-24-3p could be viewed as an independent predictor of poor overall survival and disease-free survival in patients with HBV-related hepatocellular carcinoma. *Yang et al. (2019)* suggested that miR-93-5p might be an oncogene in PC. More studies are still needed to explore how miR-93-5p functions during the life circle of prostate cancer.

Researches were demonstrating that miR-93-5p could promote gastric cancer metastasis through inactivating the Hippo signaling pathway (*Li et al., 2018*) and, on the other side, activating the STAT3 signaling pathway (*Ma et al., 2017*). *Xiang et al. (2017)* suggested that via targeting MKL-1 and STAT3 signaling pathway, miR-93-5p could inhibit the epithelial-mesenchymal transition. *Li et al. (2019)* suggested that as to high-risk HPV-positive cervical cancer, suppression of miR-93-5p might inhibit cancer cell progression by targeting BTG3. MiR-93-5p may also play an essential role via down-regulating PPARGC1A in hepatocellular carcinoma cells (*Wang et al., 2018b*).

It was reported that circulating miRNAs might be released from tumor tissues (*Endzeliņš et al., 2016*). Hence, we explored the three serum miRNAs' expression levels in PC tissues and corresponding adjacent normal tissues. Unfortunately, the result didn't achieve statistical significance. However, data of a larger sample size of unpaired tumor tissue and normal tissues from TCGA showed that miR-146a-5p and miR-93-5p were consistent with the miRNAs' expression in serum of our study, while miR-24-3p was the opposite. These suggested that the three miRNAs were closely related to PC.

Moreover, studies showed that circulating miRNAs could be embedded in exosomes or bound with Ago1-4 proteins independent from small membrane vesicles (*Simpson et al., 2009*; *Jeppesen et al., 2019*). In this study, we also identified serum miRNAs in exosomes to find the potential form of the three selected miRNAs. All the three miRNAs in serum exosomes showed high expression levels, but only miR-93-5p was presented with statistical significance in PC compared to NCs. A relatively small sample size might be partly responsible for the results.

There are still some limitations that we should consider. Firstly, more work needs to be done in the future to explore the value of the three identified miRNAs in assessing tumor dynamics. Secondly, how these miRNAs exactly play their roles in PC is not clear, and further investigations are needed.

In conclusion, we identified a three-miRNA panel for PC detection. This study displays preliminary results and will be further verified among larger samples with more statistical processes clinical application in the future. We believe that this work will serve as the basis of circulating miRNAs in clinical for the diagnosis of PC in the future, at least for the Asian population to a certain extent.

## CONCLUSION

To summarize, we have identified a three-miRNA panel (miR-146a-5p, miR-24-3p, and miR-93-5p) for the detection of PC in serum. This work can lay the foundation for clinical diagnosis of PC in the future. Still, larger cohorts of participants are needed to focus on the three miRNAs' mechanisms in PC.

## ACKNOWLEDGEMENTS

I would like to show my deepest gratitude to all volunteers involved in this study. Their understanding and dedication enable medicine to explore and advance continuously.

### Funding

This work was supported by the National Natural Science Foundation of China [Grant number: 81672400; 81702364] and the Natural Science Foundation of Jiangsu Province [Grant number: BK20171085]. The funders had no role in study design, data collection and analysis, decision to publish, or preparation of the manuscript.

### Grant Disclosures

The following grant information was disclosed by the authors:
National Natural Science Foundation of China: 81672400, 81702364.
Natural Science Foundation of Jiangsu Province:  BK20171085.

### Competing Interests

The authors declare there are no competing interests.

### Author Contributions

- Shiyu Zhang analyzed the data, prepared figures and/or tables, authored or reviewed drafts of the paper, and approved the final draft.
- Cheng Liu performed the experiments, authored or reviewed drafts of the paper, and approved the final draft.
- Xuan Zou performed the experiments, analyzed the data, prepared figures and/or tables, and approved the final draft.
- Xiangnan Geng analyzed the data, authored or reviewed drafts of the paper, and approved the final draft.
- Xin Zhou conceived and designed the experiments, prepared figures and/or tables, and approved the final draft.
- XingChen Fan analyzed the data, prepared figures and/or tables, and approved the final draft.
- Danxia Zhu and Huo Zhang performed the experiments, authored or reviewed drafts of the paper, and approved the final draft.
- Wei Zhu conceived and designed the experiments, authored or reviewed drafts of the paper, and approved the final draft.

### Human Ethics

The following information was supplied relating to ethical approvals (i.e., approving body and any reference numbers):

The First Affiliated Hospital of Nanjing Medical University and Jiangsu Cancer Hospital approved this research (Ethical Application Ref: 2016-SRFA-149).

### Ethics

The following information was supplied relating to ethical approvals (i.e., approving body and any reference numbers):

The First Affiliated Hospital of Nanjing Medical University and Jiangsu Cancer Hospital approved this research (2016-SRFA-149).

## Microarray Data Deposition

The following information was supplied regarding the deposition of microarray data:

The file "Supplemental raw data of miRNA sequence" has been uploaded.

## Data Availability

The raw measurements are available in the Supplementary Files.

## Supplemental Information

Supplemental information for this article can be found online at http://dx.doi.org/10.7717/peerj.11441#supplemental-information.

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
