# Peer review of "MicroRNA panel in serum reveals novel diagnostic biomarkers for prostate cancer"

_PeerJ, doi:10.7717/peerj.11441_

## Round 0.1 · original submission · Major Revisions

All reviewers agree that the manuscript holds the potential for publication after addressing major criticisms mentioned in the review reports.

·

Basic reporting

The article is written in a clear and appropriate English and self-contained. The provided information is sufficiently supported by literature references. As far as article structure is concerned, I think that some sections of the article contain repetitive information, for example paragraph 3.1 looks like a repetition of what already explained in section 2.1. Moreover, the methods section about the use of the miRCURY PCR Human panel (lines 121-127) in my opinion does not fit in the “study design” paragraph, I would put it either in the 2.4 section or as a separate paragraph. Figure legends are totally inadequate, they do not contain a description of the graphic elements, such as the reported bars, the sample labels ect…. Moreover, I think that figure 4 and 5 have been exchanged by mistake. In table 2 I do not understand why 4 pools are reported while only three pools are described in the text, in table 3 it is not clear what the combined column stands for.

Experimental design

The study well designed, is performed in a quite large cohort of patients and I appreciate the inclusion of training and testing phases however I think that, in order to understand value of the testing phases, a sample size calculation should be included in the methods. In the materials and methods, I would add more details about blood sample processing. Moreover, in the “external validation in public databases” paragraph I think the authors should indicate if the evaluated databases contain data about tissue of serum miRNA expression.

Validity of the findings

The study is designed to respond to a quite significant issue about prostate cancer diagnosis, however several studies are already available about the possible utility of circulating miRNAs as PC biomerkers (as stated by the authors themselves) so the authors should better argument in the discussion section in what their proposed results are outperforming compared to others. In paragraph 3.6 authors compare their proposed signature with others already available in different databases but I would consider this comparison valid only if it includes databases about circulating miRNAs.

Additional comments

The proposed study is well designed but several inaccuracies should be cleared. Moreover the authors should stress the strength of their proposed biomarkers over others already investigated.

·

Basic reporting

Zhang et al report a serum diagnostic biomarker signature of miRNA in patients with prostate cancer. The authors use primary tumor samples, bioinformatic analysis of global patients data, exosome and serum profiling to support their findings. The language used is clear with appropriate use and interpretation of the existing literature. The structure and flow of the figures and provided tables is adequate. I find the concept interesting and I would support the publication, if the authors address and expand their model as outlined in the preceding paragraphs.

Experimental design

The experimental design and methods sections are clear. The graphical abstract in figure 1 facilitates the comprehension of the experimental approach by the authors. The techniques used in this study are considered basic but reflect the original experimental question and support the conclusions.

Validity of the findings

The authors present an interesting concept identifying novel miRNA biomarkers in patients with prostate cancer. I have a couple of suggestions in order to strengthen their proposed model

1. Although the miRNA signature is clear, the authors need to experimentally and bioinformatically validate the some known target of their listed miRNAs in the same samples. An established way to screen for potential miRNA targets is the miRDB pipeline (http://mirdb.org). This will go a long way in supporting the whole signaling network affecting prostate cancer prognosis.

2. Furthermore, it would be interesting to plot Kaplan Meier curves from TCGA and other GSE datasets for these miRNA targets and correlate them with poor prognosis.

Additional comments

Zhang et al present an interesting concept for a novel biomarker identification in patients with prostate cancer. If the authors address and expand their experimental podel I would be happy to support he publication.

Reviewer 3 ·

Basic reporting

In this study, Zhang proposed experimental and bioinformatics analysis for identifying novel diagnostic biomarkers for prostate cancer. MicroRNA panel in serum reveals has been collected and they convinced some important biomarkers of prostate cancer. There are some major points that need to be addressed:

English language should be improved significantly. There are a lot of grammatical errors and typos in this manuscript. The authors should re-check and revise carefully. For example:
- We also compared diagnostic value of the 3-miRNA panel with previously reported biomarkers and verified in four public datasets.
- The diagnostic performance of the identified 3-miRNA signature show stable in the large cohorts of the three public datasets and superior than previously identified miRNA biomarkers.
- ... which could act as a non-invasive serum biomarker to help diagnosing PC.
- ...

Literature reviews are weak. There is a need to provide more literature reviews on bioinformatics-based PC analysis.

"Introduction" is short and does not emphasize the objective/motivation of the study.

Experimental design

A critical issue is that the authors only used a few sample size of data. This amount cannot be considered enough for analysis. Thus, the authors are suggested to have more data if possible.

ROC curve analysis has been used in previous works on bioinformatics such as PMID: 31362508, PMID: 31750297, and PMID: 33539511. Therefore, the authors are suggested to refer to more works in this description.

It is not clear how the authors divided the data into different sets (28, 32, and 26 patients).

The "Methodology" section should be described clearly to help to replicate the methods.

Validity of the findings

It is good to have validation data, but what are the differences in characteristics among training and validation data?

Figure legends should be described clearly. For example, in some figures, I did not understand the information in sub-figures.

The authors uploaded figures as "Tables".

The authors should compare to previous works on the same data/problem.

Additional comments

No comment

---

## Round 0.2 · Minor Revisions

One of the reviewers still suggests some revisions. Please follow all suggestions they raised in order to have the manuscript published in PeerJ.

·

Basic reporting

Zhang et al have provided an extensively revised version of their work that satisfactory addresses all the comments raised in the original revision. The manuscript is easy to follow and many inaccuracies have been corrected.

Experimental design

The newly added data are well controlled and explained.

Validity of the findings

No comments

Additional comments

Zhang et al have significantly revised their manuscript and proposed model. I support the publication of this concept.

Reviewer 3 ·

Basic reporting

No comment.

Experimental design

No comment.

Validity of the findings

No comment.

Additional comments

Thanks for addressing my comments. However, some comments still have not been addressed well. Also, some comments have been shown in the rebuttal letter, but the authors did not change in the revised manuscript. Therefore, the authors are suggested to have a minor change before acceptance.
1. More literature review related to bioinformatics analyses should be added in the "Introduction".
2. Did the authors concern about the batch effect removal when merging the data?
3. It is necessary to cite more related works to ROC analyses i.e., PMID: 31362508 and PMID: 31750297.

---

## Round 0.3 · accepted · Accept

The manuscript is now suitable for publication on PeerJ.